# Altered excitatory and inhibitory neuronal subpopulation parameters are distinctly associated with tau and amyloid in Alzheimer's disease

Kamalini G Ranasinghe[1]*, Parul Verma[2], Chang Cai[2], Xihe Xie[2], Kiwamu Kudo[2,3], Xiao Gao[2], Hannah Lerner[1], Danielle Mizuiri[2], Amelia Strom[1], Leonardo Iaccarino[1], Renaud La Joie[1], Bruce L Miller[1], Maria Luisa Gorno-Tempini[1], Katherine P Rankin[1], William J Jagust[4], Keith Vossel[1,5], Gil D Rabinovici[1,2], Ashish Raj[2], Srikantan S Nagarajan[2]

[1]Memory and Aging Center, Department of Neurology, University of California, San Francisco, San Francisco, United States; [2]Department of Radiology and Biomedical Imaging, University of California, San Francisco, San Francisco, United States; [3]Medical Imaging Business Center, Ricoh Company, Kanazawa, Japan; [4]Helen Wills Neuroscience Institute, University of California, Berkeley, Berkeley, United States; [5]Mary S. Easton Center for Alzheimer's Disease Research, Department of Neurology, David Geffen School of Medicine, University of California, Los Angeles, Los Angeles, United States

*For correspondence: kamalini.ranasinghe@ucsf.edu

## Abstract

**Background:** Neuronal- and circuit-level abnormalities of excitation and inhibition are shown to be associated with tau and amyloid-beta (Aβ) in preclinical models of Alzheimer's disease (AD). These relationships remain poorly understood in patients with AD.

**Methods:** Using empirical spectra from magnetoencephalography and computational modeling (neural mass model), we examined excitatory and inhibitory parameters of neuronal subpopulations and investigated their specific associations to regional tau and Aβ, measured by positron emission tomography, in patients with AD.

**Results:** Patients with AD showed abnormal excitatory and inhibitory time-constants and neural gains compared to age-matched controls. Increased excitatory time-constants distinctly correlated with higher tau depositions while increased inhibitory time-constants distinctly correlated with higher Aβ depositions.

**Conclusions:** Our results provide critical insights about potential mechanistic links between abnormal neural oscillations and cellular correlates of impaired excitatory and inhibitory synaptic functions associated with tau and Aβ in patients with AD.

**Funding:** This study was supported by the National Institutes of Health grants: K08AG058749 (KGR), F32AG050434-01A1 (KGR), K23 AG038357 (KAV), P50 AG023501, P01 AG19724 (BLM), P50-AG023501 (BLM and GDR), R01 AG045611 (GDR); AG034570, AG062542 (WJ); NS100440 (SSN), DC176960 (SSN), DC017091 (SSN), AG062196 (SSN); a grant from John Douglas French Alzheimer's Foundation (KAV); grants from Larry L. Hillblom Foundation: 2015-A-034-FEL (KGR), 2019-A-013-SUP (KGR); grants from the Alzheimer's Association: AARG-21-849773 (KGR); PCTRB-13-288476 (KAV), and made possible by Part the CloudTM (ETAC-09-133596); a grant from Tau Consortium (GDR and WJJ), and a gift from the S. D. Bechtel Jr. Foundation.

## Editor's evaluation

The authors explored the relationship between amyloid-β and tau deposition and neural oscillations in Alzheimer's disease (AD) by using a computational neural mass model that can generate neurophysiological power spectra comparable to EEG- or MEG-like, macroscopic brain activity assessments. This analysis demonstrates the different, frequency-specific effects of amyloid-β and tau proteins on excitation and inhibition, providing an integrated, multimodal explanation of AD pathogenesis.

## Introduction

Aggregation and accumulation of amyloid-beta (Aβ) and tau proteins are a defining feature of Alzheimer's disease (AD) pathophysiology (*Braak and Braak, 1991*). Although the mechanisms by which AD proteinopathy exerts its effects remain an area of active research, disruption of the fine balance between excitatory and inhibitory neuronal activity has emerged as a potential driver for network dysfunction contributing to cognitive deficits in AD (*Palop et al., 2006*; *Harris et al., 2020*). Preclinical AD models have demonstrated direct effects of tau and Aβ leading to impaired function in excitatory pyramidal neurons as well as inhibitory interneurons (*Palop et al., 2007*; *Hoover et al., 2010*; *Sun et al., 2012*; *Verret et al., 2012*; *Palop and Mucke, 2016*; *Zhou et al., 2017*; *Busche et al., 2019*; *Zott et al., 2019*; *Harris et al., 2020*; *Chang et al., 2021* ). In patients with AD, abnormalities in brain oscillations (*Ranasinghe et al., 2014*; *Nakamura et al., 2018*; *Maestú et al., 2019*; *Babiloni et al., 2020*; *Ranasinghe et al., 2020*), which are essentially determined by relative contributions of excitatory and inhibitory synaptic currents (*Buzsáki et al., 2012*), are a display of perturbed balance of excitation and inhibition in local circuits. However, despite the fact that clinical studies have demonstrated associations between abnormal oscillatory signatures and AD proteinopathy (*Nakamura et al., 2018*; *Smailovic et al., 2018*; *Pusil et al., 2019*; *Ranasinghe et al., 2020*;

**Table 1.** Participant demographics and clinical characteristics.

| Characteristic | Controls (N = 35) | Patients with AD (N = 20) | p[†] |
|---|---|---|---|
| Age (year) | 69.3.6 ± 8.4 | 66.3 ± 9.8 | 0.237 |
| Female sex, no. (%) | 20 (57.1) | 11 (55.0) | 0.876 |
| White, no. (%) [‡] | 30 (90.9) | 20 (100.0) | 0.282 |
| Education (year) | 18 (16–18) | 18 (16–18) | 0.855 |
| Right handedness, no. (%) | 30 (85.7) | 17 (85.0) | 0.340 |
| MMSE | 30 (29–30) | 23 (22–26) | <0.0001 |
| CDR[*] | 0 (0–0) | 0.5 (0.5–0.8) | <0.0001 |
| CDR-SOB[*] | 0 (0–0) | 3.5 (2.3–4.3) | <0.0001 |
| Age at disease onset | . | 59.4±9.39 | . |
| Disease duration | . | 6.9±2.4 | . |

Values for age, age at disease onset, and disease duration are means ± SD. Values for education, Mini Mental State Exam (MMSE), Clinical Dementia Rating (CDR), and CDR-Sum of Boxes (CDR-SOB) are medians with interquartile ranges within parentheses.

[*]Scores on the CDR range from 0 to 3 and scores on the CDR-SOB range from 0 to 18, with higher scores denoting more disability. Scores on the MMSE range from 0 to 30, with higher scores denoting better cognitive function. AD = Alzheimer's disease.

[†]Statistical tests: p values are reported from unpaired *t*-test for age, Pearson $\chi^2$ test for sex and handedness, Fisher's exact test for race, Wilcoxon–Mann–Whitney test for education, MMSE, CDR, and CDR-SOB.

[‡]Race or ethnic group was self-reported. Two control participants opted out from reporting the race.

*Ranasinghe et al., 2021*), the electrophysiological basis of aberrant excitatory and inhibitory activity of neuronal cell populations and how these relate to Aβ and tau remain largely unknown in patients with AD.

The goal of this study was to identify impaired neuronal parameters in excitatory and inhibitory neuronal subpopulations and determine their specific associations to regional Aβ and tau pathology in AD patients. We combined spectral signatures derived from magnetic field potentials via noninvasive imaging in AD patients with mathematical modeling (neural mass model, NMM) (*David and Friston, 2003*; *Raj et al., 2020*; *Verma et al., 2022*), to estimate excitatory and inhibitory neuronal parameters. Specifically, we hypothesized that abnormal regional spectral signatures in AD patients related to altered activity of excitatory and inhibitory neuronal subpopulations will be associated with tau and Aβ depositions. We combined NMM, and multimodal imaging data from: magnetoencephalography (MEG), Aβ-, and tau-positron emission tomography (PET), in a well-characterized cohort of AD patients. First, we leveraged the millisecond temporal resolution and superior spatial resolution of MEG signal to derive the oscillatory signatures of local neuronal synchrony. Next, we used a linearized NMM, which was recently described as a component of a spectral graph model, and which successfully reproduced the empirical macroscopic properties of oscillatory signatures (*Raj et al., 2020*; *Verma et al., 2022*), to derive excitatory and inhibitory parameters of local neuronal ensembles. We then examined the specific associations of altered excitatory and inhibitory neuronal subpopulation parameters and Aβ- and tau-tracer uptake patterns and how these contribute to produce the characteristic spectral changes in AD patients.

## Materials and methods
### Participants
Twenty patients with AD (diagnostic criteria for probable AD or mild cognitive impairment due to AD) (*Albert et al., 2011*; *McKhann et al., 2011*; *Jack et al., 2018*) and 35 age-matched controls were included in this study (*Table 1*). Each participant underwent a complete clinical history, physical examination, neuropsychological evaluation, brain magnetic resonance imaging (MRI), and a 10-min session of resting MEG. All AD patients underwent PET with tau-specific radiotracer, [18]F-flortaucipir and Aβ-specific radiotracer, [11]C-PIB. Twelve AD patients in this study cohort overlapped with our previous multimodal imaging investigation of long-range synchrony assay (*Ranasinghe et al., 2020*). All participants were recruited from research cohorts at the University of California San Francisco-Alzheimer's Diesease Research Center(UCSF-ADRC). Informed consent was obtained from all participants and the study was approved by the Institutional Review Board (IRB) at UCSF (UCSF-IRB 10-02245).

### Clinical assessments and MEG, PET, and MRI acquisition and analyses
AD patients were assessed via MMSE and a standard battery of neuropsychological tests. All participants were assessed via a structured caregiver interview to determine the Clinical Dementia Rating (CDR) (Appendix 2).

MEG scans were acquired on a whole-head biomagnetometer system (275 axial gradiometers; MISL, Coquitlam, British Columbia, Canada) for 5–10 min, following the same protocols described previously (*Ranasinghe et al., 2020*). Tomographic reconstructions of source-space data were done using a continuous 60second data epoch, an individualized head model based on structural MRI, and a frequency optimized adaptive spatial filtering technique implemented in the Neurodynamic Utility Toolbox for MEG (NUTMEG; http://nutmeg.berkeley.edu). We derived the regional power spectra based on Desikan–Killiany atlas parcellations for the 68 cortical regions depicting neocortex and allocortex, the latter including the entorhinal cortex. Regional power spectra were derived from FFT and then converted to dB scale for the following frequency bands: 2–7 Hz, delta–theta; 8–12 Hz, alpha; 13–35 Hz, beta; and 1–35 Hz, broad-band (Appendix 2).

Flortaucipir and PIB-PET acquisitions were done based on the same protocols detailed previously (*Schöll et al., 2016*). Standardized uptake value ratios (SUVRs) were created using Freesurfer-defined cerebellar gray matter for PIB-PET. For [18]F-flortaucipir, Freesurfer segmentation was combined with the SUIT template to include inferior cerebellum voxels, avoiding contamination from off-target binding in the dorsal cerebellum (Appendix 2).

## Mathematical modeling and parameter estimation

We used a linearized NMM (**Raj et al., 2020**; **Verma et al., 2022**) to estimate excitatory and inhibitory neuronal subpopulation parameters. In this regional model, for every region $k$ ($k$ varies from 1 to $N$ and $N$ is the total number of regions) based on the Desikan–Killiany parcellation, the regional population signal is modeled as the sum of excitatory signals $x_e(t)$ and inhibitory signals $x_i(t)$. Both excitatory and inhibitory signal dynamics consist of a decay of the individual signals with a fixed neural gain, incoming signals from populations that alternate between the excitatory and inhibitory signals, and input Gaussian white noise. The equations for the excitatory and inhibitory signals for every region are the following:

$$\frac{dx_e(t)}{dt} = -\frac{f_e(t)}{\tau_e} \star \left( g_{ee}x_e(t) - g_{ei}f_i(t) \star x_i(t) \right) + p(t)$$

$$\frac{dx_i(t)}{dt} = -\frac{f_i(t)}{\tau_i} \star \left( g_{ii}x_i(t) + g_{ei}f_e(t) \star x_e(t) \right) + p(t)$$

The symbols used in the equations are as following: * stands for convolution; parameters $g_{ee}$, $g_{ii}$, and $g_{ei}$ are neural gains for the excitatory, inhibitory, and alternating populations, respectively; $\tau_e$ and $\tau_i$ are time-constants of excitatory and inhibitory populations, respectively; $p(t)$ is the input Gaussian white noise; $f_e(t)$ and $f_i(t)$ are Gamma-shaped ensemble average neural impulse response functions (see Appendix 2 for step-by-step details). The parameters, $g_{ee}$, $g_{ii}$, $\tau_e$, and $\tau_i$ were estimated for each region-of-interest (ROI) and parameter $g_{ei}$ was fixed at 1. The excitatory and inhibitory time-constant parameters characterize the duration of the neural responses (modeled by a Gamma-shaped function) in each neuronal subpopulation, respectively. It also characterizes the rate at which a local signal dissipates in absence of other inputs. A lower time-constant indicates a faster rate of change in signals while a higher time-constant indicates a slower rate. The excitatory and inhibitory gain parameters correspond to the postsynaptic gain on the impulse response function of each neuronal subpopulation, respectively. Each region's spectrum was modeled using the above equations, and the power spectral density was generated for frequencies 1–35 Hz. The goodness of fit of the model was estimated by calculating the Pearson's correlation coefficient between the simulated model power spectra and the empirical source localized MEG spectra for frequencies 1–35 Hz. This goodness of fit value was used to estimate the model parameters. Parameter optimization was done using the basin hopping global optimization algorithm in Python (**Wales and Doye, 1997**). The model parameter values and bounds were specified as: 17, 5, and 30 ms, respectively, for initial, lower-boundary, and upper-boundary, for $\tau_e$ and $\tau_i$ ; 0.5, 0.1, and 10, respectively, for initial, lower-boundary, and upper-boundary, for $g_{ee}$ and $g_{ii}$. The hyperparameters of the algorithm which included the number of iterations, temperature, and step size were set at 2000, 0.1, and 4, respectively. If any of the parameters was hitting the specified bounds, parameter optimization was repeated with a step size of 6 for that specific ROI. Finally the set of parameters which led to the highest Pearson's correlation coefficient was chosen. The cost function for this optimization was negative of Pearson's correlation coefficient between the source localized MEG spectra in dB scale and the model power spectral density in dB scale as well. This procedure was performed for every ROI for every subject.

## Statistical analyses

Statistical tests were performed using SAS software (SAS9.4; SAS Institute, Cary, NC). To compare the demographics and clinical characteristics between controls and patients with AD, we used unpaired $t$-tests for age, Pearson $\chi^2$ test for sex and handedness, Fisher's exact test for race, Wilcoxon–Mann–Whitney test for education, MMSE, CDR, and CDR-SOB.

We used a one-way analysis of variance (ANOVA) to compare the broad-band power spectra , and a two-way ANOVA to compare across the three frequency bands, delta–theta, alpha, and beta, between controls and patients. Each model included a repeated measures design to incorporate the 68 cortical ROIs per subject. Post hoc comparisons were derived from comparing least-squares means with the adjustment of multiple comparisons using Tukey–Kramer test. The regional patterns of spectral power distributions incorporated unpaired $t$-tests at regional level and thresholded with 10% false discovery rate.

To compare the neuronal parameters between the controls and patients we used a linear mixed-effects model (PROC MIXED) with repeated measures design to incorporate the multiple ROIs per

subject. We reported the estimated least-squares means and the statistical differences of least-squares means based on unpaired *t*-tests between patients and controls.

We ran mixed effects models to examine the associations between tau- and Aβ-tracer uptake and excitatory and inhibitory neuronal parameters derived from NMM. The predictor variables of models included the flortaucipir (tau) SUVR and $^{11}$C-PIB (Aβ) SUVR, at each ROI in patients with AD. The dependent variables included the *z*-score measures depicting the change of each neuronal parameter in patients, based on age-matched control cohort. A seperate mixed model was used for each parameter including the neuronal time-constants, $\tau_e$ and $\tau_i$, and neural gains, $g_{ee}$ and $g_{ii}$. Each model included a repeated measures design to incorporate the 68 ROIs per subject and modeled the heterogeneity in residual variances at each ROI. Mixed models to examine the associations between average scaling difference between the MEG spectra and the model output did not show any significant relationships.

We ran separate mixed effect models where the dependent variable included the *z*-score measures depicting the change of spectral power in patients, based on age-matched control cohort, within (1) broad-band spectrum (1–35 Hz), (2) delta–theta spectrum (2–7 Hz), (3) alpha spectrum (8–12 Hz), and (4) beta spectrum (13–35 Hz). Each model included a repeated measures design and modeled the heterogeneity in residual variances at each ROI.

As our analyses on NMM parameters and AD proteinopathy revealed that inhibitory and excitatory neuronal time-constants are associated with Aβ and tau accumulations, respectively, we next utilized the PROC MIXED procedure in SAS to perform a mediation analysis (*Bauer et al., 2006*) to examine the role of altered time-constants in mediating the characteristic spectral changes associated with Aβ and tau in AD. Specifically, we examined whether Aβ associated increased spectral power is mediated by increased inhibitory time-constants ($\tau_i$) and whether tau associated reduced spectral power is mediated by increased excitatory time-constants ($\tau_e$). The mediation models included spectral power (in each frequency band) as dependent variable and protein accumulation (Aβ or tau) and time-constant parameter ($\tau_i$)($\tau_e$) as predictor variables. The mediation models estimated the overall association between a given frequency band oscillation and Aβ or tau (which is equivalent to the associations derived from the mixed models described in the previous section), and then determined the time-constant mediated effect (effect that is dependent on time-constant abnormality) and the direct effect (effect that is independent of time-constant abnormality). The mediation analyses examined the following effects: (1) direct and $\tau_i$ mediated effects of Aβ on delta–theta; (2) direct and $\tau_i$ mediated effects of Aβ on alpha and beta; (3) direct and $\tau_e$ mediated effects of tau on alpha and beta.

## Results

On average, the patients were mild to moderately impaired with a mean MMSE score of 22.8 ± 4.5 (MMSE range: 22–26), mean CDR of 0.72 ± 0.47 (CDR range: 0.5–0.8), and mean CDR-SOB of 3.8 ± 2.5, with characteristic cognitive deficits (*Table 1*; *Appendix 1—table 1*).

### Regional spectral changes in AD: increased delta–theta and reduced alpha and beta

Patients with AD showed a clear leftward shift in their power spectra when compared to age-matched controls. Specifically, AD patients showed a reduced spectral power within alpha (CI, 58.04–59.85 dB, 60.33–61.69 dB, AD and controls, respectively) and beta (CI, 53.16–54.11B, 56.03–56.75 dB, AD and controls, respectively) but increased power within delta–theta bands (60.14–61.79 dB, 57.60–58.85 dB, AD and controls, respectively) (*Figure 1A, B*). A direct region-wise comparison showed a frontal predominant spatial distribution for the spectral power increase within delta–theta and a posterior predominant distribution for the spectral power reduction in alpha and beta, in patients with AD (*Figure 1C*; *Appendix 1—figure 1*). These results demonstrate the frequency-specific and region-dependent characteristics of oscillatory abnormalities in AD.

### Estimated NMM parameters demonstrate altered excitatory and inhibitory subpopulation activity

We used a linear NMM, capable of reproducing spectral properties of neural activity, to predict the empirical spectra at regional level (i.e., 68 cortical regions) in patients with AD and controls. NMM

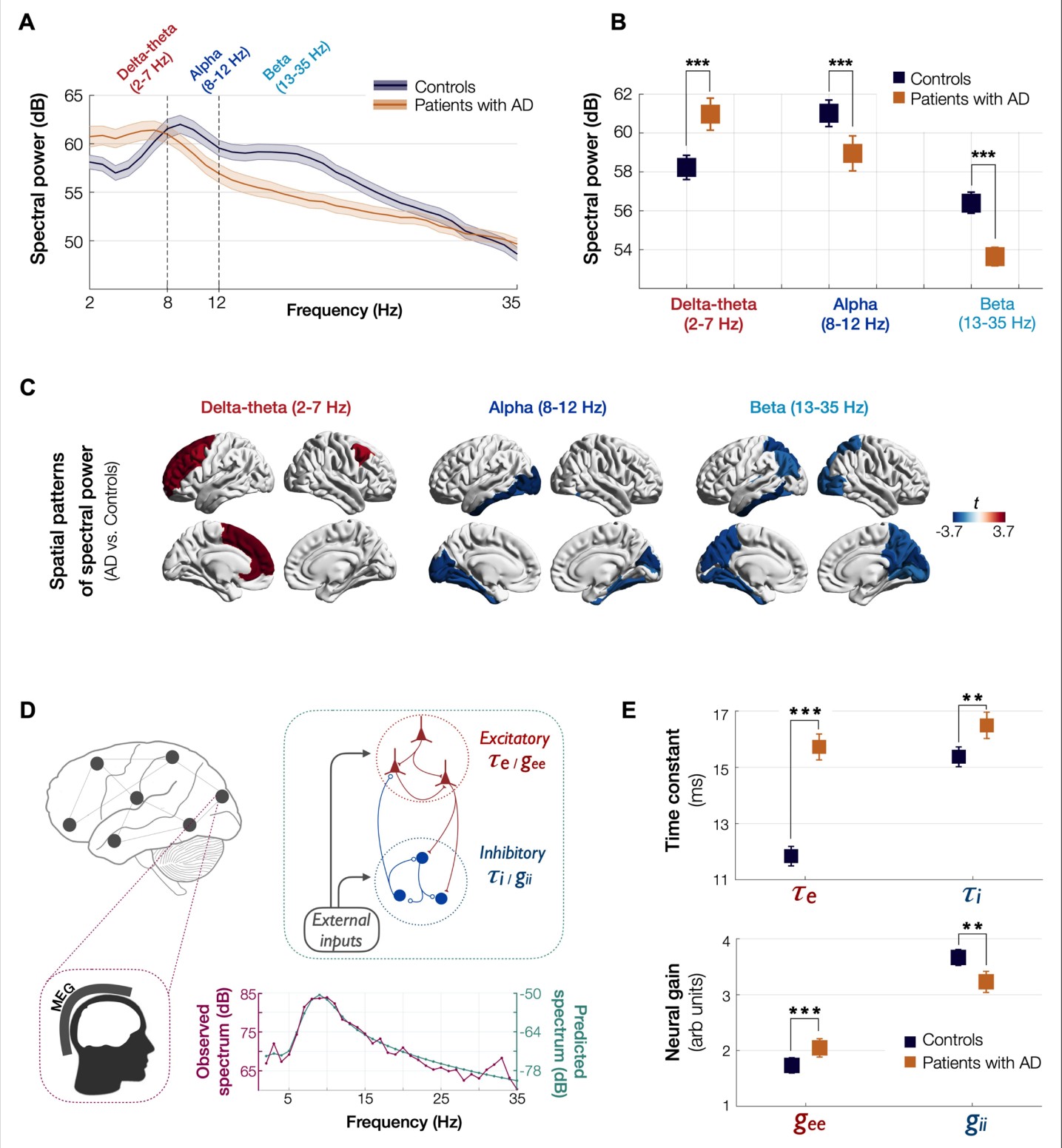

**Figure 1.** Spectral power changes and altered excitatory and inhibitory neuronal subpopulation parameters in patients with AD. Patients with AD showed higher delta–theta (2–7 Hz) spectral power and lacked a clear alpha peak (8–12 Hz) as opposed to controls (**A**). A two-way ANOVA comparing patients and controls showed significantly higher spectral power within delta–theta frequency band and showed significantly lower spectral power within alpha and beta (13–35 Hz) bands, in AD patients (**B**). The markers depict the least-square means, and the error bars depict the 95% confidence intervals. Regional patterns of spectral power changes in patients with AD showed increased delta–theta power is predominant in the frontal regions and reduced

*Figure 1 continued on next page*

*Figure 1 continued*

alpha and beta spectral power is predominant in the temporoparietal and occipital cortices (**C**). Images show the *t*-values from statistical comparison of regional data, based on DK atlas parcellations, and thresholded at FDR 10%. Schematic representation of the linear neural mass model (NMM) (**D**), where the NMM represents local assemblies of excitatory and inhibitory neurons at each region of interest (ROI) lumped into linear systems. External inputs and outputs are gated through both excitatory and inhibitory neurons. The recurrent architecture of the two pools within a local area is captured by the neuronal time-constants, $\tau_e$ and $\tau_i$, and neural gain terms, $g_{ee}$ and $g_{ii}$, indicating the loops created by recurrents within excitatory, inhibitory, and cross-populations. At each ROI, the model delivers these parameters as it predicts the broad-band spectrum (1–35 Hz) optimized to the empirical spectrum derived from MEG. Patients with AD showed significantly increased neuronal time-constants, $\tau_e$ and $\tau_i$ compared to age-matched controls (**E**). Patients with AD also showed increased excitatory neural gains ($g_{ee}$) and reduced inhibitory neural gains ($g_{ii}$) than controls (c). The markers and error bars depict the least-square means and 95% confidence intervals. Abbreviations: AD, Alzheimer's disease; MEG, magnetoencephalography.

predicted four parameters for neuronal populations: excitatory time-constant ($\tau_e$), inhibitory time-constant ($\tau_i$), excitatory neural gain ($g_{ee}$), and inhibitory neural gain ($g_{ii}$). Specifically, in each subject, and for each cortical region, the NMM parameters were estimated based on the best fit (highest Pearson correlation coefficient) between observed MEG power spectrum and the predicted NMM spectrum (*Figure 1D*; *Appendix 1—figure 2*). Statistical mixed models with repeated measures demonstrated that AD patients have significantly increased time-constant parameters of excitatory and inhibitory neurons ($\tau_e$ and $\tau_i$) than controls (*Figure 1E*; $\tau_e$: CI, 15.27–16.19, 11.49–12.18; p < 0.0001; $\tau_i$: CI, 16.03–16.96, 15.02–15.73; p = 0.0002, AD and controls, respectively). Furthermore, AD patients showed increased $g_{ee}$ and reduced $g_{ii}$ compared to controls indicating abnormal neural gains in both

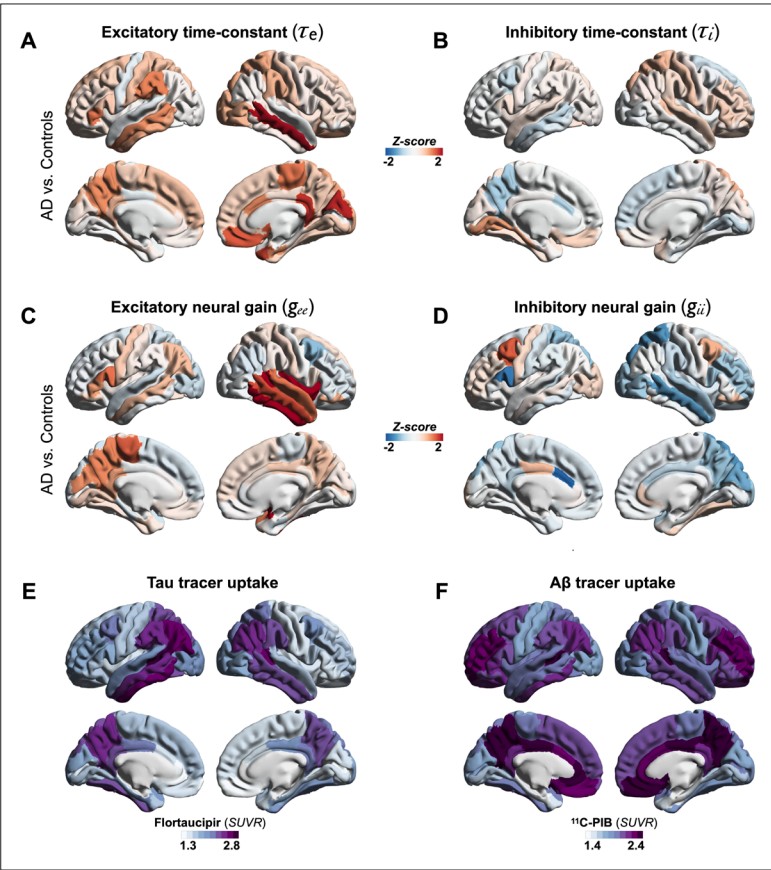

**Figure 2.** Regional patterns of neuronal subpopulation parameters and protein tracer uptakes in patients with AD. Subplots A–D depict the regional differences (*z*-scores) for excitatory time-constant (**A**), inhibitory time-constant (**B**), excitatory gain (**C**), and inhibitory gain (**D**) parameters in AD patients when compared to age-matched controls. Subplots E and F depict the average regional patterns of flortaucipir standardized uptake value ratio (SUVR) (**E**) and $^{11}$C-PIB SUVR (**F**) for patients with AD showing high flortaucipir retention in temporal lobe, posterior and lateral parietal regions, and high $^{11}$C-PIB retention in bilateral frontal and posterior parietal cortices. Abbreviations: AD, Alzheimer's disease; Aβ, amyloid-beta.

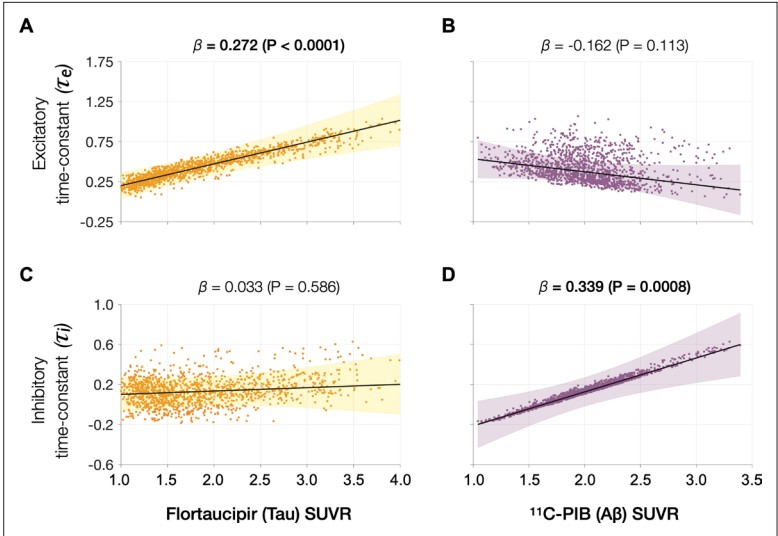

**Figure 3.** Associations between tau- and Aβ-tracer uptake and excitatory and inhibitory neuronal time-constants in patients with AD. Increased time-constants showed distinct associations with tau and Aβ in AD patients. Increased excitatory time-constant ($\tau_e$) was positively correlated with tau, but not with Aβ (**A, B**). Increased inhibitory time-constant ($\tau_i$) was positively correlated with Aβ, but not with tau (**C, D**). Subplots A–D indicate the model estimates from linear mixed-effects models predicting the changes (z-scores) in each neuronal parameter from flortaucipir (tau) standardized uptake value ratio (SUVR) and $^{11}$C-PIB (Aβ) SUVR, in patients with AD. The fits depicting tau predictions were computed at the average SUVR of Aβ (1.99), and the fits depicting Aβ were computed at average SUVR of tau (1.64). The scatter plots indicate the predicted values from each model incorporating a repeated measures design. Abbreviations: AD, Alzheimer's disease; Aβ, amyloid-beta.

excitatory and inhibitory subpopulations (**Figure 1E**; $g_{ee}$: CI, 1.88–2.21, 1.59–1.87; p = 0.0005; $g_{ii}$: CI, 3.04–3.42, 3.52–3.81; p = 0.0003, AD and controls, respectively). The regional patterns of increased excitatory time-constants and neural gains showed the highest changes in a spatial pattern involving thetemporal lobe and the precuneus (**Figure 2A, C**). Increased inhibitory time-constants showed a distributed spatial pattern involving frontal and parietal cortices (**Figure 2B**), while the reduced inhibitory neural gains showed the highest reductions in the right temporal and posterior parietal regions (**Figure 2C**).

## Tau and Aβ distinctly modulate excitatory and inhibitory time-constants, respectively

Next, we examined the functional associations of model parameters with flortaucipir (tau) and $^{11}$C-PiB (Aβ) uptake patterns (**Figure 2E, F**). linear mixed-effects models showed that increased $\tau_e$ was correlated with higher tau-tracer uptake, while increased $\tau_i$ was correlated with higher Aβ-tracer uptake (**Figure 3A, D**; $\tau_e$: tau, t = 4.11, p < 0.0001; $\tau_i$: Aβ, t = 3.38, p = 0.0008). In contrast, there were no correlations between $\tau_e$ and Aβ-tracer uptake and between $\tau_i$ and tau-tracer uptake (**Figure 3B, C**; $\tau_e$: Aβ, t = −1.59, p = 0.1131; $\tau_i$: tau, t = 0.54, p = 0.5863). In contrast to time-constant associations, altered neural gains did not show statistically significant associations to either flortaucipir or $^{11}$C-PiB uptakes (**Appendix 1—figure 3**). Distinctive association of tau with excitatory time-constants and Aβ with inhibitory time-constants may support the hypothesis of distinct roles of tau- and Aβ-mediated pathomechanisms on excitatory and inhibitory synaptic functions.

## Spectral changes associated with tau and Aβ are partially mediated by altered excitatory and inhibitory time-constants

Next, we tested the hypothesis that effects of tau and Aβ on the frequency-specific spectral power changes would be mediated by their unique modulatory effects on $\tau_e$ and $\tau_i$, respectively. To this end, we first demonstrated the specific relationships between frequency-specific spectral changes and regional tracer uptake (flortaucipir and $^{11}$C-PiB). Consistent with previous reports (**Canuet et al., 2015**; **Nakamura et al., 2018**; **Pusil et al., 2019**; **Ranasinghe et al., 2020**), linear mixed model

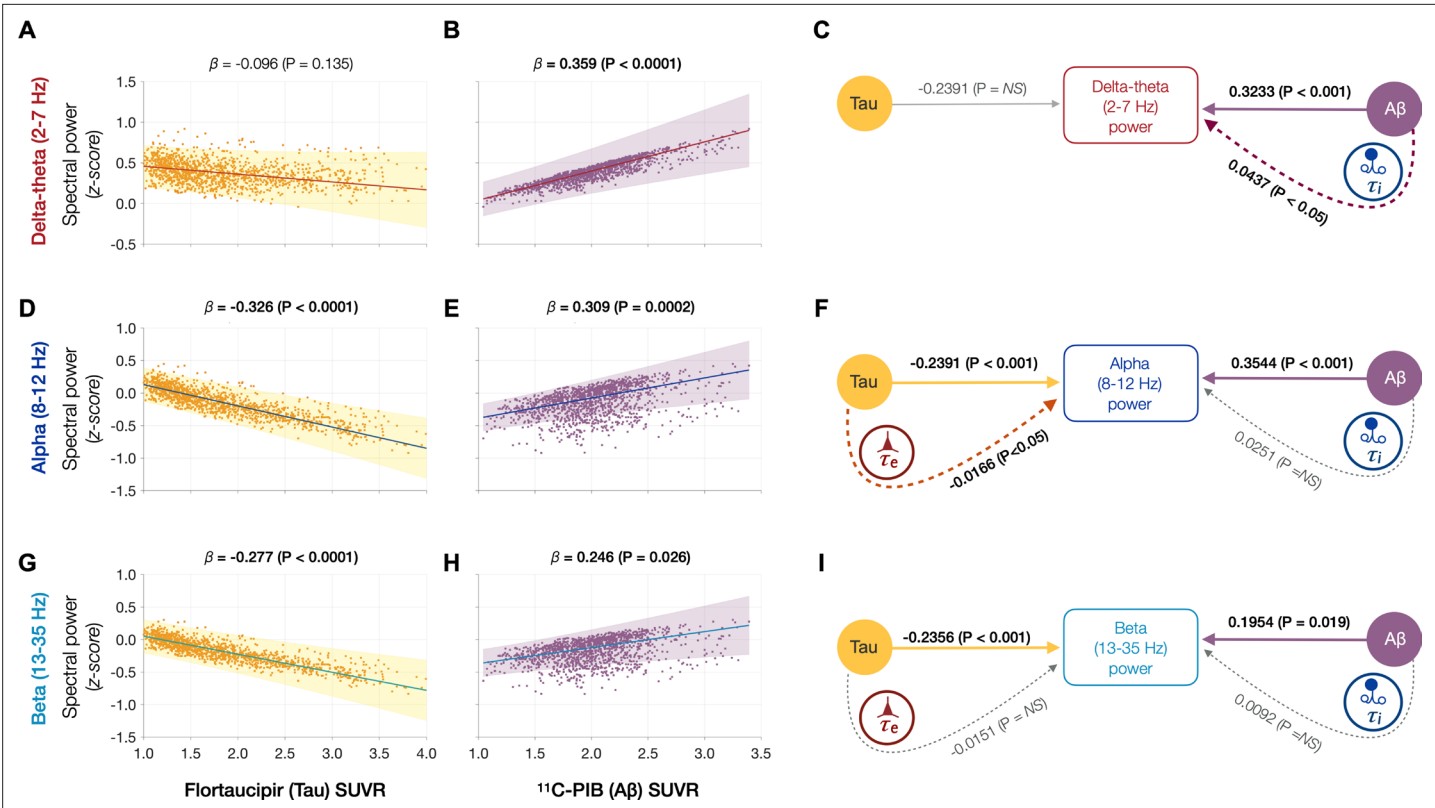

**Figure 4.** Frequency-specific spectral power modulations of tau and Aβ are partially mediated via increased excitatory ($\tau_e$) and inhibitory ($\tau_i$) time-constants. Associations between tau- and Aβ-tracer uptake and spectral power changes in patients with AD are depicted in subplots A, B, D, E, G, and H. Tau was not associated with the delta–theta (2–7 Hz) spectral changes (**A**), while it was positively modulated by Aβ (**B**). Both alpha (8–12 Hz) and beta (13–35 Hz) spectra showed significant negative associations with tau (**D, G**) and significant positive associations with Aβ (**E, H**). Subplots indicate the model estimates from linear mixed-effects analyses predicting the spectral power changes from flortaucipir (tau) SUVR and ¹¹C-PIB (Aβ) SUVR, for patients with AD. The fits depicting tau predictions were computed at the average SUVR of Aβ (1.99), while the fits depicting Aβ were computed at average SUVR of tau (1.64). The scatter plots indicate the predicted values from each model incorporating a repeated measures design to account for 68 regions per subject. Subplots C, F, and I depict mediation models to examine the direct effects of tau and Aβ, and the effects mediated through excitatory ($\tau_e$) and inhibitory ($\tau_i$) time-constants on different frequency bands. Delta–theta power increases were significantly affected by Aβ and were partially mediated through the effect of Aβ on inhibitory ($\tau_i$) time-constant (**C**). Alpha power reductions were affected by tau and a small, but a significant fraction of this effect was mediated through the effect of tau on excitatory ($\tau_e$) time-constant (**F**). Beta power reductions were significantly affected by tau, although there was no statistically significant effect mediated through the effect of tau on excitatory ($\tau_e$) time-constant (**I**). Aβ effects on alpha and beta spectral changes were only direct effects with no statistically significant effects mediated through altered inhibitory ($\tau_i$) time-constants. Abbreviations: AD, Alzheimer's disease; Aβ, amyloid-beta; SUVR, standardized uptake value ratio.

analyses showed that associations of tau and Aβ on the power spectrum were frequency specific. For example, delta–theta was only associated with Aβ (positive correlation) and showed no associations to tau (*Figure 4A, B*). In contrast, alpha and beta power spectra showed significant associations to both tau and Aβ, where higher tau reduced spectral power and higher Aβ increased spectral power (*Figure 4D, E, G, H*). Including regional cortical atrophy as a covariate into the models did not influence these relationships, indicating that spectral changes are robust to neuronal loss (*Appendix 1— figure 4*). In summary, delta–theta power was uniquely associated with Aβ while reduced alpha and beta spectral power were the result of a dual modulation by tau and Aβ with a net negative modulatory effect from tau.

Next, we used a mediation analysis to examine whether the distinct effects of tau and Aβ on frequency-specific spectral changes are mediated via altered $\tau_e$ and $\tau_i$ , respectively. The mediation analyses specifically examined: (1) the direct and the $\tau_i$ mediated effects of Aβ on delta–theta power; (2) the direct and the $\tau_e$ mediated effects of tau on alpha and beta power; and (3) the direct and the $\tau_i$ mediated effects of Aβ on alpha and beta power. We found that Aβ modulation of delta–theta power was significantly mediated through $\tau_i$ in addition to direct modulation (*Figure 4C*). We also found that

tau modulation of alpha power was significantly mediated through $\tau_e$ in addition to the direct modulation (*Figure 4F*), whereas Aβ modulation of alpha power was only through a direct effect. Tau as well as Aβ modulation of beta power occurred only though direct effects (*Figure 4I*). Collectively, $\tau_e$ and $\tau_i$ partially mediated the effects of AD proteinopathy toward signature spectral changes observed in AD.

## Discussion

This is the first study, in patients with AD, showing quantitative links between altered neuronal subpopulation dynamics of excitatory and inhibitory function, and abnormal accumulations of tau and Aβ. We combined electrophysiology, molecular imaging, and NMM model, to examine the excitatory and inhibitory parameters of regional neural subpopulations in patients with AD and how these relate to tau and Aβ depositions. AD patients showed abnormal excitatory and inhibitory neuronal parameters compared to controls and with distinct associations to tau and Aβ where higher tau correlated with increased excitatory time-constants and higher Aβ correlated with increased inhibitory time-constants. Furthermore, the frequency-specific associations of spectral changes to tau and Aβ were partially mediated by increased excitatory and inhibitory time-constants, respectively. Collectively, our findings demonstrate distinct functional consequences of tau and Aβ at the level of circuits where cellular and molecular changes of AD pathophysiology possibly converge, and provide a rationale to identify potential mechanisms of excitation–inhibition imbalance, hyperexcitability, and abnormal neural synchronization in AD patients that could help guide future clinical trials.

### Abnormal excitatory and inhibitory time-constants represent differential functional consequences of AD pathophysiology at circuit level

Unlike invasive basic science approaches that can be designed to examine causal relationships, clinical investigations for the most part are limited to examine associative relationships. Nonetheless, the associative links from clinical studies provide essential building blocks to link the findings from preclinical models to clinical manifestations in patients. NMM is currently by far the most sophisticated tool to investigate circuit function at the level of excitatory and inhibitory neuronal subpopulations in the human brain using noninvasive imaging modalities. The finding that excitatory and inhibitory time-constant abnormalities are uniquely correlated with higher tau and Aβ, respectively, draws a few key insights in the context of our evolving understanding of AD pathobiology.

The distinctive association of higher tau accumulations to increased excitatory time-constants which indicate aberrant excitatory function within local ensembles of neuronal subpopulations, is consistent with multiple lines of evidence suggesting how tau affects excitatory function of neural circuits. For example, neuropathological studies in human patients with AD detailing the morphology and location of cells that accumulate tau and degenerate indicate an increased vulnerability of excitatory neurons to tau-related pathomechanisms (*Hyman et al., 1984*; *Braak and Braak, 1991*). In basic science studies, mice expressing mutant human tau demonstrate impaired synaptic transmission of glutamate leading to reduced firing of pyramidal neurons (*Hoover et al., 2010*; *Fu et al., 2017*; *Fu et al., 2019*) while tau reduction in transgenic mice produce an overall decrease in baseline excitatory neuronal activity and modulate the inhibitory neuronal activity leading to reduced network excitation (*Chang et al., 2021*). The collective insight from these observations indicates a relative vulnerability of excitatory function in neural networks to tau and a resulting network hypoactivity (*Harris et al., 2020*). Two key findings from the current study are consistent with this discernment which include: (1) excitatory neuronal parameters uniquely associated with increased tau depositions; (2) reduced oscillatory activity of alpha band associated with higher tau being partially mediated by abnormal excitatory time-constants. Although these findings do not exclude the possibility of tau directly altering firing patterns of inhibitory neurons (*Chang et al., 2021*), they support the hypothesis that the effects of tau pathophysiology within local networks manifest as excitatory function deficits.

In contrast to intracellular aggregates of tau, accumulation of Aβ is extracellular (*Braak and Braak, 1991*; *Nagy et al., 1995*). AD basic science models have demonstrated a range of Aβ-associated pathomechanisms that ranges from toxic effects of different Aβ forms affecting both excitatory and inhibitory synaptic functions (*Meyer-Luehmann et al., 2008*; *Busche et al., 2012*; *Busche et al., 2015*; *Zott et al., 2019*). A potential means by which Aβ leads to network dysfunction in animal

models of AD is abnormal hyperactivity in cortical and hippocampal neurons (*Palop and Mucke, 2016*). Compelling evidence from AD transgenic mice indicate impaired inhibitory synaptic function as a contributory cause for Aβ-related neuronal hyperactivity (*Busche et al., 2008*; *Busche et al., 2012*; *Verret et al., 2012*). Our findings draw remarkable parallels to these basic science observations by showing unique associations between inhibitory time-constant abnormalities and higher Aβ tracer uptake. It is important to reiterate that the current findings indicate an overall inhibitory functional deficit at the level of local networks which in turn may be contributed by abnormal inhibitory as well as excitatory deficits at cellular level. Basic science experiments indeed have identified reduced inhibitory interneuron activity as well as aberrant glutamate transmission as potential underlying causes of network hyperactivity in AD transgenic mice (*Busche et al., 2008*; *Verret et al., 2012*; *Zott et al., 2019*).

Collectively, findings from this clinical imaging investigation, together with comparable basic science evidence, help bridge a crucial gap between circuit- and cellular-level abnormalities in AD. A key finding from preclinical AD models is that cellular-level changes associated with tau and Aβ produces a combined functional consequence of altered excitatory–inhibitory balance in neural networks (*Palop and Mucke, 2016*; *Harris et al., 2020*; *Chang et al., 2021*; *Maestú et al., 2021*). The emerging picture from basic science models of AD also suggest that abnormally increased neuronal activity associated with Aβ most likely dominate during preclinical stage of the disease (*Zott et al., 2019*), whereas the firing suppression of tau will become predominant at later stages (*Busche et al., 2019*). The current results highlight these distinct roles played by tau and Aβ in network dysfunction and suggest that neurophysiological markers are sensitive indices to pursue each pathway, conceivably along different disease stages. Future studies delineating the mechanistic relationships between increased excitatory and inhibitory time-constants and network hyperexcitability are crucial to understand how tau and Aβ impair excitatory–inhibitory balance along the biological progression of AD.

Although we found significant impairments in both excitatory and inhibitory gain parameters in AD patients, these did not show significant associations with tau and Aβ. This result maybe explained in part by the relative smaller effect sizes of gain parameters (compared to time-constants). Another possible explanation may be related to the type of molecular form associated with pathophysiological effects. In both tau and Aβ, not only that the soluble molecular forms are important mediators of neurotoxicity but also their effects predominate during the preclinical stages of the disease (*Busche, 2019*; *Zott et al., 2019*). However, PET tracer uptake represents mostly the deposited nonsoluble forms of protein accumulations. As such it is possible that abnormal neural gains may represent an

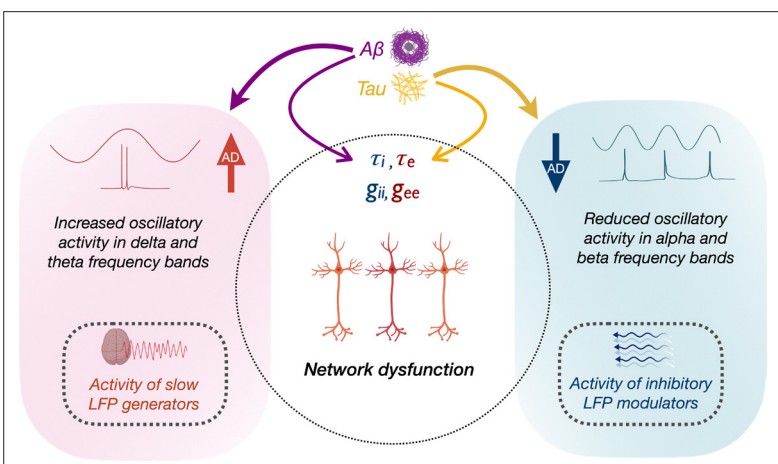

**Figure 5.** Schematic representation of the modulation of frequency-specific neuronal activity by Aβ and tau, and associated network dysfunction in AD. Tau and Aβ have distinct effects on excitatory and inhibitory neuronal parameters of local neuronal subpopulations, as well as alpha, beta and delta–theta oscillatory changes in AD. Positive modulation of delta/theta rhythms by Aβ (left panel), may potentially contribute to a status of increased network activity. Alpha/beta, on the other hand, reflects activity of inhibitory regulation of network activity (right panel). Negative modulation of alpha/beta oscillations by tau may therefore contribute to a status of less regulated network activity.

early effect of soluble neurotoxins, while abnormal time-constants may represent dynamic effects of network changes indicatingprogressive pathophysiological events.

## Frequency-specific spectral changes may indicate distinct processes leading to network dysfunction in AD

How the opposing phenomena of Aβ- and tau-associated abnormal hyper- and hypoactivity of neurons lead to a status of abnormal network dysfunction remains a conundrum. In *Figure 5*, we speculate the possible interactions of molecular and oscillatory mechanisms that could lead to network dysfunction. Although a unifying principle governing the physiology of rhythmic oscillations remains obscure, a commonly accepted principle is that oscillations regulate the top-down processing of local neuronal firing and facilitate long-range interactions (*Uhlhaas et al., 2009*). Low-frequency delta–theta and mid-frequency alpha and beta oscillations employ diverse physiological mechanisms determined by different ionic currents (*Wang, 2010*) and have distinct functional roles (*Engel et al., 2001*). The prominent view in the current literature is that delta–theta oscillations are positive top-down modulators of local neural activity whereas the power of alpha and beta exert an inhibitory modulation of irrelevant neuronal activity thus reducing the neural noise (*Klimesch, 1999*). We posit that higher delta–theta power associated with increased Aβ therefore may predispose a dysregulated increase of local firing (*Figure 5*, left panel). In patients with epilepsy increased focal and generalized slow waves are characteristic features during the interictal period, although the mechanistic relationship between network hyperexcitability and slow oscillations remains unknown. In patients with temporal lobe epilepsy higher incidence of slow waves are associated with greater volume loss in medial temporal lobe structures (*Cascino et al., 1996*; *Cendes et al., 1996*). Although it is possible that Aβ-associated unknown mechanisms may relate to both neuronal death and hyperexcitability, the relationship between Aβ and atrophy in AD is weak as the two phenomena are widely apart in temporal evolution and anatomical distribution.

Our results are also consistent with opposing modulations from tau and Aβ, and an overall net effect of tau resulting in reduced neuronal activity (*Harris et al., 2020*). For example, alpha and beta oscillations were positively modulated by Aβ and negatively modulated by tau, albeit a stronger net negative effect with reduced alpha and beta power. Because alpha oscillations are considered as inhibitory gain controllers of local circuits (*Klimesch et al., 2007*; *Lorincz et al., 2009*), we posit that a net reduction of alpha may yet again be favorable for a status of dysregulated network activity (*Figure 5*, right panel). Collectively, the multimodal neuroimaging in AD patients in the current study demonstrates how positive oscillatory modulators (delta–theta) are associated with Aβ, while negative oscillatory modulators (alpha) are associated with tau. Together, these findings suggest that the paradoxical relationship between tau, Aβ, and network dysfunction, could be better understood by the frequency-specific nature of oscillatory abnormalities. Future studies are warranted to further delineate the contributions from excitatory and inhibitory subpopulation functions toward network hyperexcitability and their interplay with oscillatory spectral changes. Overall, this framework offers a new perspective for evaluating and understanding future efforts in network stabilizing therapies.

## Oscillatory spectral changes reflect dynamic functional deficits in AD

Previous MEG/EEG studies have further shown that abnormal neurophysiological indices also represent dynamic changes of AD pathophysiology. Individuals with preclinical stages of AD and APOE4 carriers who carry an elevated risk of developing AD show increased alpha power and synchrony in select regions including medial frontal and posterior parietal cortices (*Cuesta et al., 2015*; *Nakamura et al., 2018*; *Pusil et al., 2019*), whereas in patients with AD dementia syndrome the alpha power and long-range synchrony are reduced (*Sami et al., 2018*; *Ranasinghe et al., 2020*). For example, a study using Aβ-PET in cognitively normal controls and MCI patients showed that Aβ-positive cognitively normal participants have higher alpha power than Aβ-negative cognitively normal whereas, Aβ-positive MCI patients had reduced alpha spectral power compared to cognitively normal regardless of Aβ status (*Nakamura et al., 2018*). These data support the hypothesis that hyperactive effects of Aβ dominate in the preclinical and prodromal stages of AD with subsequent effects of tau and the complex synergy of both proteins leading to hypoactivity in neural circuits. The dynamic association of the dual proteinopathy is a crucial factor in developing new disease modifying drugs for AD for they may not only explain that the dominance of tau as a possible reason for the relative lack of efficiency

in anti-Aβ trials, but also indicate the importance of targeting the dual modulation of tau and Aβ. Frequency-specific oscillatory signatures provide attractive biomarkers to track the dynamics of AD pathophysiology in the next generation of AD therapeutic trials.

## NMMs in AD research

Although quantitative electrophysiological assays of neural oscillations provide the most direct measures of neuronal and synaptic function in the human brain (*Buzsáki et al., 2012*), it is only by combining these fine spectral details with mathematical modeling (*David and Friston, 2003*) that we can delineate neuronal level details from noninvasive neuroimaging in human subjects. Current NMMs are capable of depicting more realistic forms of synaptic and network interactions and have proved especially successful in simulating the pathological alterations of distinct excitatory and inhibitory neurons in diseases such as AD (*de Haan et al., 2017*). Furthermore, using a nonlinear NMM a recent study also showed consistent findings of a positive relationship between inhibitory time-constant and higher Aβ suggesting an association between Aβ accumulation and spectral slowing (*Stefanovski et al., 2019*). A key difference is the use of a nonlinear form of NMM model by Stefanovski et al. as opposed to the linear version in our study. While linearizing is a simplification of the detailed under-lying biophysics, recent comparisons among different models demonstrate that linear models suffi-ciently capture neuroimaging data with higher accuracy compared to nonlinear models. In addition, the small set of model parameters and the closed-form solution in the frequency domain in our model makes the parameter inference more tractable compared to nonlinear versions of NMM. Indeed, we were able to show accurate fits to empirical spectra capturing the empirical peak frequency as well as the frequency fall-off. We do not however, observe bifurcation points and other bistable behaviors that can be observed in a nonlinear NMM. Notwithstanding the differences, these studies collec-tively illustrate an important role of NMM applications in expressing abnormalities in excitatory and inhibitory neuronal parameters which may help unify the electrophysiological findings from clinical AD populations and from AD transgenic mice. Future experiments extending the NMM applications to global network properties in addition to local neuronal synchrony may elucidate the relationships between altered neuronal parameters and hierarchical network organizations in AD.

## Limitations

Our findings should be considered in the context of the following limitations. First, it is important to point out that any computational model may not perfectly capture the complex dynamics of struc-ture–function coupling in the human brain. Nonetheless, our model has the advantage of using only a few parameters which were interpretable in terms of the underlying biophysics. Second, PET signal represents the deposited proteins and is mostly insensitive to soluble forms of proteins, although basic science models suggest that soluble oligomers are concentrated around deposited proteins (*Busche et al., 2008*). Another limitation includes the known off-target binding of flortaucipir in the basal ganglia, choroid plexus, and the meninges. However, these off-target sources of signal are unlikely to have driven our results because the flortaucipir increases in the current study were seen in brain areas remote from these sites. While the current study was limited to examine the pathophys-iological consequences on network properties in AD patients, it is equally important to understand the same phenomena in normal aging. It is also noteworthy that functional changes associated with AD pathophysiology are dynamic along the biological progression of the disease and will be best investigated in future longitudinal study designs. Finally, the current sample size limited the ability to establish a natural history of the excitatory and inhibitory neuronal parameters, which will be the focus of future investigations.

## Acknowledgements

We would like to thank all the study participants and their families for their generous support to our research. We would also like to acknowledge Avid Radiopharmaceuticals for enabling the use of the 18F-flortaucipir tracer by providing the precursor. This study was supported by the National Insti-tutes of Health grants: K08AG058749 (KGR), F32AG050434-01A1 (KGR), K23 AG038357 (KAV), P50 AG023501, P01 AG19724 (BLM), P50-AG023501 (BLM and GDR), R01 AG045611 (GDR); AG034570, AG062542 (WJ); NS100440 (SSN), DC176960 (SSN), DC017091 (SSN), AG062196 (SSN); a grant from John Douglas French Alzheimer's Foundation (KAV); grants from Larry L Hillblom Foundation:

2015-A-034-FEL (KGR); 2019-A-013-SUP (KGR); grants from the Alzheimer's Association: AARG-21-849773 (KGR); PCTRB-13-288476 and made possible by Part the CloudTM (ETAC-09-133596) (KAV); a grant from Tau Consortium (GDR and WJJ), and a gift from the S D Bechtel Jr. Foundation.

## Additional information

### Competing interests

Kiwamu Kudo: is an employee of Ricoh Company, Ltd. The author declares that no other competing interests exist. Bruce L Miller: serves as Medical Director for the John Douglas French Foundation; Scientific Director for the Tau Consortium; Director/Medical Advisory Board of the Larry L. Hillblom Foundation; and Scientific Advisory Board Member for the National Institute for Health Research Cambridge Biomedical Research Centre and its subunit, the Biomedical Research Unit in Dementia, UK. The other authors declare that no competing interests exist.

### Funding

| Funder | Grant reference number | Author |
| --- | --- | --- |
| National Institute on Aging | K08AG058749 | Kamalini G Ranasinghe |
| National Institute on Aging | K23 AG038357 | Keith Vossel |
| National Institutes of Health | | Bruce L Miller<br>William J Jagust<br>Gil Rabinovici<br>Ashish Raj<br>Srikantan Nagarajan |
| Alzheimer's Association | PCTRB-13-288476; AARG-21-849773 | Kamalini G Ranasinghe<br>Keith Vossel |
| Larry L. Hillblom Foundation | 2015-A-034-FEL | Kamalini G Ranasinghe |
| National Institutes of Health | F32AG050434-01A1 | Kamalini G Ranasinghe |
| National Institutes of Health | P50-AG023501 | Bruce L Miller<br>Gil D Rabinovici |
| National Institutes of Health | P01 AG19724 | Bruce L Miller |
| National Institutes of Health | R01 AG045611 | Gil D Rabinovici |
| National Institutes of Health | AG034570 | William J Jagust |
| National Institutes of Health | AG062542 | William J Jagust |
| National Institutes of Health | NS100440 | Srikantan S Nagarajan |
| National Institutes of Health | DC176960 | Srikantan S Nagarajan |
| National Institutes of Health | DC017091 | Srikantan S Nagarajan |
| National Institutes of Health | AG062196 | Srikantan S Nagarajan |
| John Douglas French Alzheimer's Foundation | | Keith Vossel |
| Larry L. Hillblom Foundation | 2019-A-013-SUP | Kamalini G Ranasinghe |

| Funder | Grant reference number | Author |
|---|---|---|
| Tau Consortium | | Gil D Rabinovici<br>William J Jagust |

The funders had no role in study design, data collection, and interpretation, or the decision to submit the work for publication.

## Author contributions

Kamalini G Ranasinghe, Gil D Rabinovici, Ashish Raj, Srikantan S Nagarajan, Conceptualization, Data curation, Formal analysis, Funding acquisition, Investigation, Methodology, Project administration, Resources, Software, Supervision, Validation, Visualization, Writing – original draft, Writing – review and editing; Parul Verma, Formal analysis, Methodology, Validation, Writing – original draft, Writing – review and editing; Chang Cai, Xihe Xie, Kiwamu Kudo, Leonardo Iaccarino, Renaud La Joie, Formal analysis; Xiao Gao, Hannah Lerner, Amelia Strom, Maria Luisa Gorno-Tempini, Data curation; Danielle Mizuiri, Data curation, Project administration; Bruce L Miller, Conceptualization, Funding acquisition, Resources, Supervision, Writing – review and editing; Katherine P Rankin, Conceptualization, Resources, Supervision, Writing – review and editing; William J Jagust, Conceptualization, Data curation, Formal analysis, Methodology, Resources, Supervision, Writing – original draft, Writing – review and editing; Keith Vossel, Conceptualization, Data curation, Formal analysis, Funding acquisition, Investigation, Methodology, Resources, Supervision, Writing – original draft, Writing – review and editing

## Author ORCIDs

Kamalini G Ranasinghe http://orcid.org/0000-0002-4217-8785
Parul Verma http://orcid.org/0000-0001-9956-4954
Kiwamu Kudo http://orcid.org/0000-0002-5732-7229
Renaud La Joie http://orcid.org/0000-0003-2581-8100
William J Jagust http://orcid.org/0000-0002-4458-113X
Ashish Raj http://orcid.org/0000-0003-2414-2444
Srikantan S Nagarajan http://orcid.org/0000-0001-7209-3857

## Ethics

Informed consent was obtained from all participants and the study was approved by the Institutional Review Board (IRB) at UCSF (UCSF-IRB 10-02245).

## Decision letter and Author response

Decision letter https://doi.org/10.7554/eLife.77850.sa1
Author response https://doi.org/10.7554/eLife.77850.sa2

# Additional files

## Supplementary files
• Transparent reporting form

## Data availability

All data associated with this study are present in the paper or in the supplementary material. Anonymized subject data will be shared on request from qualified investigators for the purposes of replicating procedures and results, and for other noncommercial research purposes within the limits of participants' consent. Correspondence and material requests should be addressed to kamalini.ranasinghe@ucsf.edu.

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

# Appendix 1

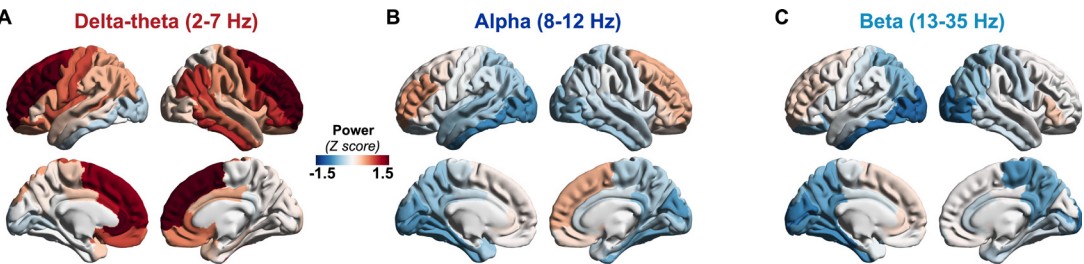

**Appendix 1—figure 1.** Regional patterns of spectral power change in patients with AD. Regional change in spectral power distributions within delta–theta (2–7 Hz), alpha (8–12 Hz), and beta (13–35 Hz) frequency bands in patients with AD (**A–C**) as depicted in z-scores estimated based on age-matched controls. Patients with AD showed frontal predominant increases in delta–theta spectral power and posterior predominant reductions in alpha and beta spectral power. Over the frontal regions AD patients showed a trend toward increased alpha and beta spectral power. Abbreviation: AD, Alzheimer's disease.

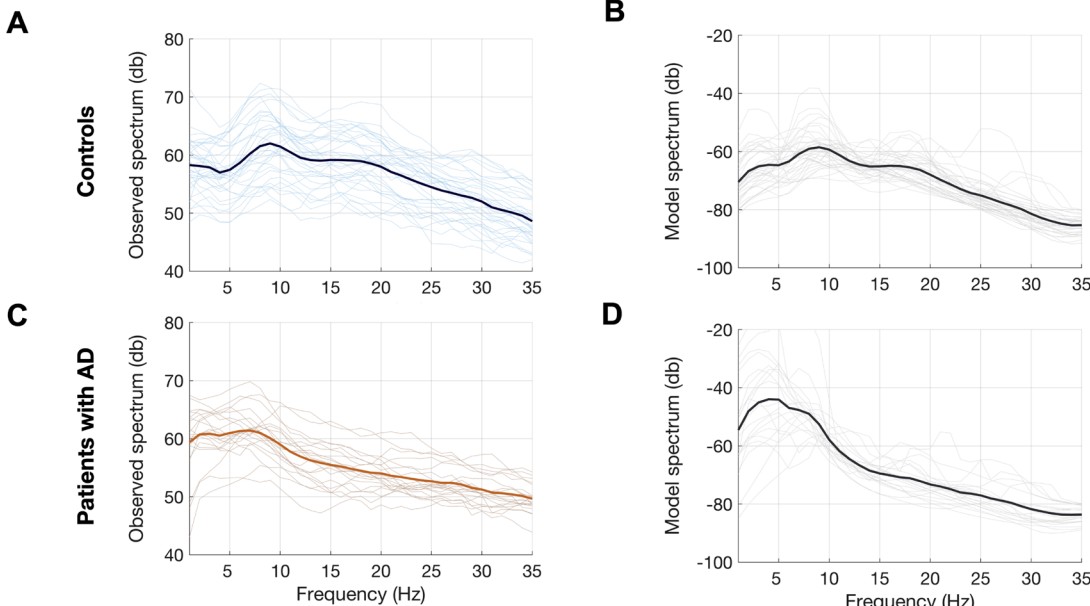

**Appendix 1—figure 2.** Observed and predicted power spectra in patients with AD and age-matched controls. Observed and model predicted spectra for each participant in the age-matched controls (**A, B**) and patients with AD (**C, D**). Each individual line depicts the average spectrum for a given subject across 68 cortical ROIs. The dark lines depict group averages. The observed spectra were derived from the source space reconstructed MEG time-series data. The model spectra were generated from the linear neural mass model with optimized neuronal parameters for time-constants (excitatory, $\tau_e$ and inhibitory, $\tau_i$) and neural gains (excitatory, $g_{ee}$ and inhibitory, $g_{ii}$) to predict the broad-band spectrum (1–35 Hz) optimized to the empirical spectrum derived from MEG. Abbreviations: AD, Alzheimer's disease; MEG, magnetoencephalography; ROI, regions-of-interest.

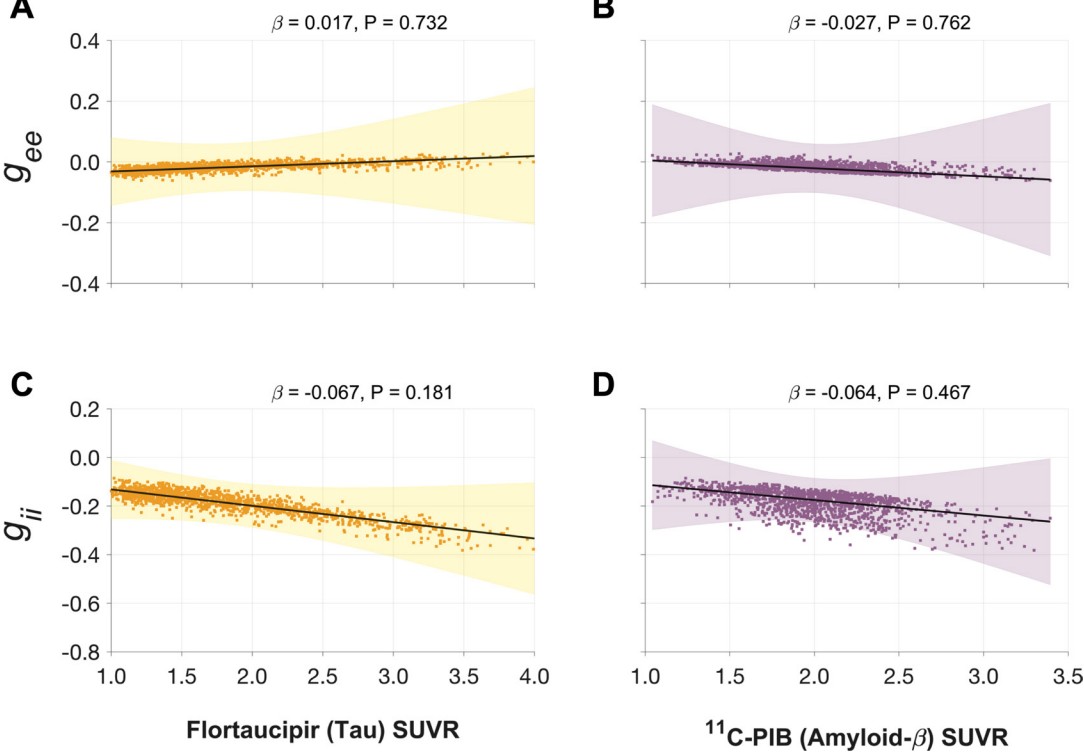

**Appendix 1—figure 3.** Associations between tau- and Aβ-tracer uptake and neuronal gain parameters in patients with AD. Altered gain parameters did not show significant associations with tau and Aβ in AD patients. Subplots A–D indicate the model estimates from linear mixed-effects models predicting the changes (z-scores) in each neuronal parameter from flortaucipir (tau) SUVR and $^{11}$C-PIB (Aβ) SUVR, in patients with AD. The fits depicting tau predictions were computed at the average SUVR of Aβ (1.99), and the fits depicting Aβ were computed at average SUVR of tau (1.64). The scatter plots indicate the predicted values from each model incorporating a repeated measures design. Abbreviations: AD, Alzheimer's disease; Aβ, amyloid-beta; $g_{ee}$, excitatory gain; $g_{ii}$, inhibitory gain; MEG, magnetoencephalography; SUVR, standardized uptake value ratio.

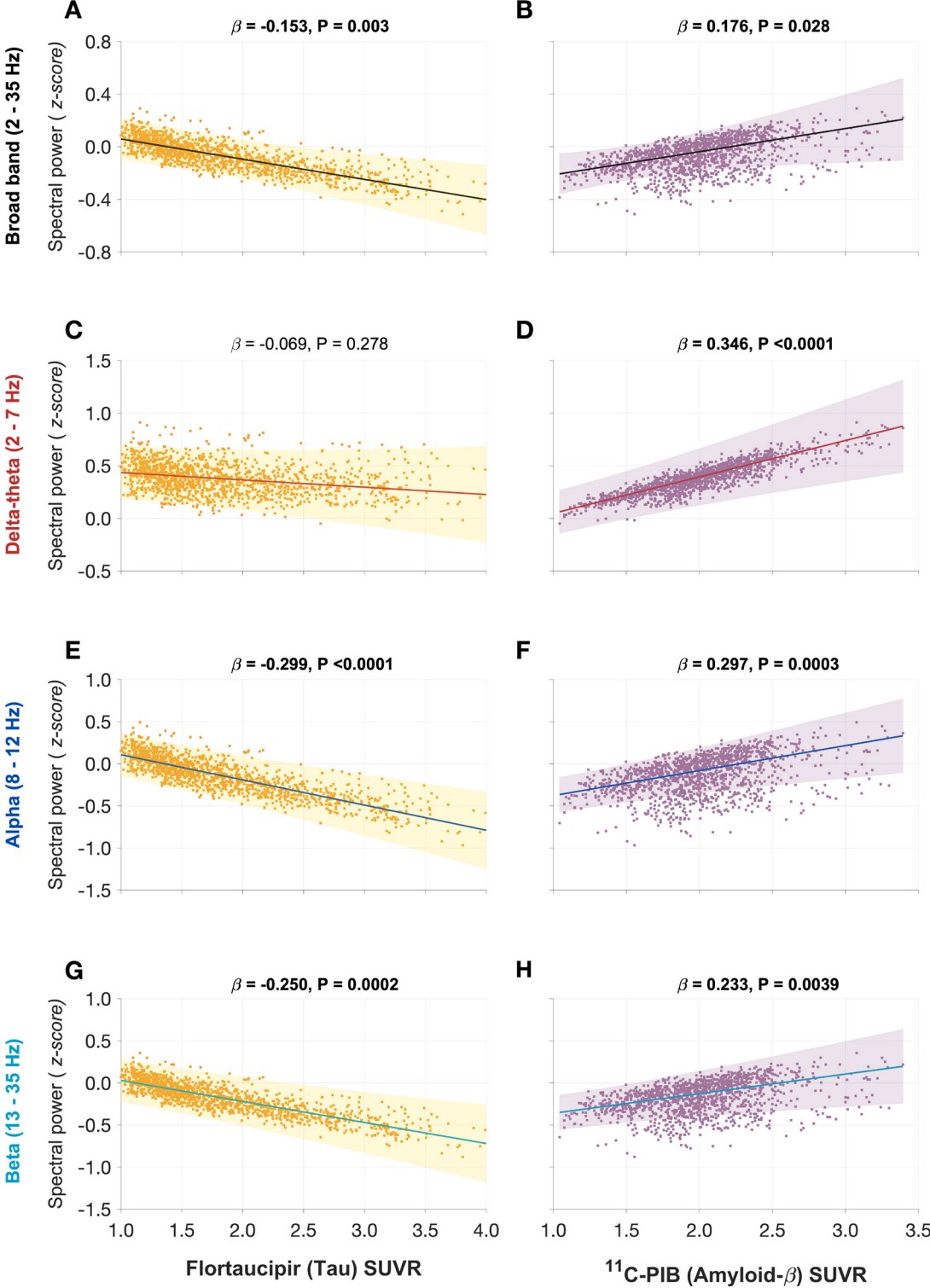

**Appendix 1—figure 4.** Associations between spectral power changes and tau- and Aβ-tracer uptake after correcting for regional atrophy. Tau showed a significant negative association (**A**), while Aβ showed a significant positive association (**B**), with the broad-band power spectrum (2–35 Hz). These effects were distinct within each frequency-specific spectrum. Tau was not associated with the delta–theta (2–7 Hz) spectral changes (**C**), while it was positively modulated by Aβ (**D**). Both alpha (8–12 Hz) and beta (13–35 Hz) spectra showed significant negative associations with tau and significant positive associations with Aβ (**E–H**). Each subplot indicates the estimates from linear mixed-effects models predicting the spectral power changes from flortaucipir (tau) SUVR and 11C-
*Appendix 1—figure 4 continued on next page*

*Appendix 1—figure 4 continued*
PIB (Aβ) SUVR, after including the additional covariate of cortical atrophy in each ROI, in patients with AD. The fits depicting tau predictions were computed at the average SUVR of Aβ (1.99), while the fits depicting Aβ were computed at average SUVR of tau (1.64), each at the average *w*-score of cortical volume (−0.62). The scatter plots indicate the predicted values from each model incorporating a repeated measures design to account for 68 regions per subject. *Z*-scores for spectral power values were calculated based on the normal control cohort. Abbreviations: AD, Alzheimer's disease; Aβ, amyloid-beta; SUVR, standardized uptake value.

**Appendix 1—table 1.** Neuropsychological test performance in patients with AD.

| Variable | Test score (mean ± SD) |
|---|---|
| Episodic memory function | |
| Visual free recall (Benson 10 min) | 4.9 ± 3.2 |
| Short delay verbal memory (CVLT 30 s) | 3.9 ± 2.4 |
| Verbal free recall (CVLT 10 min) | 2.3 ± 2.9 |
| Executive function and working memory | |
| Design fluency | 6.4 ± 2.9 |
| Information processing speed (Stroop color naming) | 48.2 ± 17.6 |
| Cognitive control (Stroop Inhibition) | 21.3 ± 12.9 |
| Verbal working memory (Digit span forward) | 5.3 ± 1.33 |
| Attention (Digit span backward) | 3.8 ± 1.2 |
| Set shifting (Modified trails – speed) | 0.2 ± 0.2 |
| Verbal learning (CVLT total score) | 17.9 ± 6.6 |
| Language function | |
| Reading irregular words | 5.6 ± 0.7 |
| Syntax comprehension | 3.9 ± 1.2 |
| Verbal agility | 4.6 ± 1.2 |
| Boston Naming Test | 12.3 ± 3.1 |
| Lexical fluency (D words/1 min) | 10.8 ± 5.0 |
| Category fluency (animals/1 min) | 11.4 ± 5.0 |
| Repetition | 3.3 ± 1.4 |
| Visuospatial function | |
| Face discrimination (CATS – face matching) | 10.8 ± 1.7 |
| Visuoconstruction (Benson copy) | 12.8 ± 4.2 |
| Location discrimination (VOSP number location) | 7.5 ± 2.5 |
| *Calculations* | 3.4 ± 1.4 |
| *Emotion naming* (CATS – affect matching) | 12.7 ± 1.0 |

CVLT = California Verbal Learning Test containing nine items. CATS = Comprehensive Affect Testing System. VOSP = Visual Object and Space Perception.

## Appendix 2

### Supplementary methods

#### Resting state MEG data acquisition

Each subject underwent MEG recording on a whole-head biomagnetometer system consisting of 275 axial gradiometers (MISL, Coquitlam, British Columbia, Canada), for 5–10 min. Three fiducial coils including nasion, left and right preauricular points were placed to localize the position of head relative to sensor array, and later coregistered to each individual's respective MRI to generate an individualized head shape. Data collection was optimized to minimize within-session head movements and to keep it below 0.5 cm. 5–10 min of continuous recording was collected from each subject while lying supine and awake with eyes closed (sampling rate: 600 Hz). We selected a 60-s (1 min) continuous segment with minimal artifacts (minimal excessive scatter at signal amplitude <10 pT), for each subject, for analysis. The study protocol required the participant to be interactive with the investigator and be awake at the beginning of the data collection. Spectral analysis of each MEG recording and whenever avaiable, a the simultaneously collected scalp EEG recording were examined to confirm that the 60-s data epoch represented awake, eyes closed resting state for each participant. Artifact detection was confirmed by visual inspection of sensor data and channels with excessive noise within individual subjects were removed prior to analysis.

#### Source space reconstruction of MEG data and spectral power estimation

Tomographic reconstructions of the MEG data were generated using a head model based on each participant's structural MRI. Spatiotemporal estimates of neural sources were generated using a time–frequency optimized adaptive spatial filtering technique implemented in the Neurodynamic Utility Toolbox for MEG (NUTMEG; https://nutmeg.berkeley.edu/). Tomographic volume of source locations (voxels) was computed through an adaptive spatial filter (10-mm lead field) that weights each location relative to the signal of the MEG sensors (*Dalal et al., 2008*; *Dalal et al., 2011*). The source space reconstruction approach provided amplitude estimations at each voxel derived through the linear combination of spatial weighting matrix with the sensor data matrix (*Dalal et al., 2008*). A high-resolution anatomical MRI was obtained for each subject (see below) and was spatially normalized to the Montreal Neurological Institute (MNI) template brain using the SPM software (http://www.fil.ion.ucl.ac.uk/spm), with the resulting parameters being applied to each individual subject's source space reconstruction within the NUTMEG pipeline (*Dalal et al., 2011*).

To prepare for source localization, all MEG sensor locations were coregistered to each subject's anatomical MRI scans. The lead field (forward model) for each subject was calculated in NUTMEG using a multiple local-spheres head model (three-orientation lead field) and an 8-mm voxel grid which generated more than 5000 dipole sources, all sources were normalized to have a norm of 1. The MEG recordings were projected into source space using a beamformer spatial filter. Source estimates tend to have a bias towards superficial currents and the estimates are more error-prone when we approach subcortical regions, therefore, only the sources belonging to the 68 cortical regions were selected for further analyses. Specifically, all dipole sources were labeled based on the Desikan–Killiany parcellations, then sources within a 10-mm radial distance to the centroid of each brain region were extracted for each region. In this study, we examined the broad-band (1–35 Hz) and also the regional power spectra of three frequency bands: 2–7 Hz delta–theta band, 8–12 Hz alpha band, and 13–35 Hz beta band. The low-frequency oscillatory band in our design spanned across the conventional delta (2–4 Hz) and theta (4–8 Hz) oscillatory band. This approach was chosen to capture the full range of low-frequency oscillatory activity described in human neurophysiology (*Jacobs, 2014*; *Goyal et al., 2020*), which includes the complete window where increased spectral signature is observed in patients with AD. Power spectra were derived by applying FFT on the time-course data and then converted to dB scale.

#### Mathematical modeling and parameter estimation

We used an NMM (*David and Friston, 2003*; *Moran et al., 2013*; *Hartoyo et al., 2020*) based on an analytical and linearized version published previously (*Raj et al., 2020*; *Verma et al., 2022*) for estimation of regional model parameters. In this model, for every region $k$, where $k$ varies from 1 to $N$ and $N$ is the total number of regions based on the Desikan–Killiany parcellation the regional population signal is modeled as the sum of excitatory signals $x_e(t)$ and inhibitory signals $x_i(t)$ (*Figure 1D*).

Both excitatory and inhibitory signal dynamics consist of a decay of the individual signals with a fixed neural gain, incoming signals from populations that alternate between the excitatory and inhibitory signals, and input Gaussian white noise. The equations for the excitatory and inhibitory signals for every region are the following:

$$\frac{dx_e(t)}{dt} = -\frac{f_e(t)}{\tau_e} \star \left( g_{ee} x_e(t) - g_{ei} f_i(t) \star x_i(t) \right) + p(t)$$

$$\frac{dx_i(t)}{dt} = -\frac{f_i(t)}{\tau_i} \star \left( g_{ii} x_i(t) + g_{ei} f_e(t) \star x_e(t) \right) + p(t) \tag{1}$$

where * stands for convolution, parameters $g_{ee}$, $g_{ii}$, and $g_{ei}$ are neural gains for the excitatory, inhibitory, and alternating populations, respectively, $\tau_e$ and $\tau_i$ are characteristic time-constants of the excitatory and inhibitory populations, respectively, $p(t)$ is the input Gaussian white noise, and $f_e(t)$ and $f_i(t)$ are Gamma-shaped ensemble average neural impulse response functions written as following:

$$f_e(t) = \frac{t}{\tau_e^2} e^{\frac{-t}{\tau_e}} \tag{2}$$

$$f_i(t) = \frac{t}{\tau_i^2} e^{\frac{-t}{\tau_i}} \tag{3}$$

Since these are linear equations, the closed-form solution of $x_e(t)$ and $x_i(t)$ can be obtained in the Fourier domain as $X_e(\omega)$ and $X_i(\omega)$ respectively, where $\omega$ is the frequency, by taking a Fourier transform of *Equations 1 and 2* as the following:

$$j\omega X_e(\omega) = -\frac{F_e(\omega)}{\tau_e} \left( g_{ee} X_e(\omega) - g_{ei} F_i(\omega) X_i(\omega) \right) + P(\omega) \tag{4}$$

$$j\omega X_i(\omega) = -\frac{F_i(\omega)}{\tau_i} \left( g_{ii} X_i(\omega) + g_{ei} F_e(\omega) X_e(\omega) \right) + P(\omega) \tag{5}$$

where $j$ is the imaginary unit, $P(\omega)$ is the Fourier transform of $p(t)$, and $F_e(\omega)$ and $F_i(\omega)$ are written as the following:

$$F_e(\omega) = \frac{\frac{1}{\tau_e^2}}{\left( j\omega + \frac{1}{\tau_e} \right)^2} \tag{6}$$

$$F_i(\omega) = \frac{\frac{1}{\tau_i^2}}{\left( j\omega + \frac{1}{\tau_i} \right)^2} \tag{7}$$

Solving *Equations 6 and 7* yields the following:

$$X_e(\omega) = \frac{\left( 1 + \frac{\frac{g_{ei}}{\tau_e} F_e(\omega) F_i(\omega)}{j\omega + \frac{g_{ii}}{\tau_i} F_i(\omega)} \right) P(\omega)}{j\omega + \frac{g_{ee}}{\tau_e} F_e(\omega) + \frac{(g_{ei} F_e(\omega) F_i(\omega))^2}{\tau_e \tau_i \left( j\omega + \frac{g_{ii}}{\tau_i} F_i(\omega) \right)}} \tag{8}$$

$$X_i(\omega) = \frac{\left( 1 - \frac{\frac{g_{ei}}{\tau_i} F_e(\omega) F_i(\omega)}{j\omega + \frac{g_{ee}}{\tau_e} F_e(\omega)} \right) P(\omega)}{j\omega + \frac{g_{ii}}{\tau_i} F_i(\omega) + \frac{(g_{ei} F_e(\omega) F_i(\omega))^2}{\tau_e \tau_i \left( j\omega + \frac{g_{ee}}{\tau_e} F_e(\omega) \right)}} \tag{9}$$

Thus, $X_e(\omega)$ and $X_i(\omega)$ can be written as $H_e(\omega) P(\omega)$ and $H_i(\omega) P(\omega)$, respectively, where $H_e(\omega)$ and $H_i(\omega)$ are the transfer functions and $P(\omega)$ is the driving function. The simulated spectra $X(\omega) = X_e(\omega) + X_i(\omega) = (H_e(\omega) + H_i(\omega)) P(\omega)$, and the power spectral density is estimated as $E\left( |X(\omega)|^2 \right)$, where $E$ is the expectation. Since the driving function $P(\omega)$ is Gaussian noise which has a flat power spectrum, $E\left( |X(\omega)|^2 \right) \propto |H_e(\omega) + H_i(\omega)|^2$. Finally, it is converted to dB scale by calculating $10\log_{10}\left( |H_e(\omega) + H_i(\omega)|^2 \right)$.

The parameters, $g_{ee}$, $g_{ii}$, $\tau_e$, and $\tau_i$ were estimated for each ROI and parameter $g_{ei}$ was fixed at 1. Each region's spectrum was modeled using the above equations, and the power spectral density was generated for frequencies 1–35 Hz. The goodness of fit of the model was estimated by calculating the Pearson's correlation coefficient between the simulated model power spectra and the empirical source localized MEG spectra for frequencies 1–35 Hz. This goodness of fit value was used to estimate the model parameters. Parameter optimization was done using the basin hopping global optimization algorithm in Python (*Wales and Doye, 1997*). The model parameter values and bounds were specified as: 17, 5, and 30 ms, respectively, for initial, upper-boundary, and lower-boundary, for $\tau_e$ and $\tau_i$ ; 0.5, 0.1, and 10, respectively, for initial, upper-boundary, and lower-boundary, for $g_{ee}$ and $g_{ii}$. The hyperparameters of the algorithm which included the number of iterations, temperature, and step size were set at 2000, 0.1, and 4, respectively. If any of the parameters was hitting the specified bounds, parameter optimization was repeated with a step size of 6 for that specific ROI, and finally the set of parameters which led to a higher Pearson's correlation coefficient was chosen. The cost function for this optimization was negative of Pearson's correlation coefficient between the source localized MEG spectra in dB scale and the model power spectral density in dB scale as well. This procedure was performed for every ROI of every subject.

In order to examine the effects of model parameters on excitatory and inhibitory activity, $X_e(\omega)/X_i(\omega)$ was calculated while varying each of the parameters $g_{ee}$, $g_{ii}$, $\tau_e$, and $\tau_i$ one-by-one, keeping others fixed at their estimated mean values calculated for the control cohort. This exploration demonstrated the complex dependency of $X_e(\omega)/X_i(\omega)$ on parameters which varied in a frequency-dependent manner. The complex predictions from $g_{ee}$ and $g_{ii}$ illustrated their control effect on the decay terms in *Equations 1 and 2*. For instance, when $g_{ee}$ is increased, $x_e(t)$ decays sooner whereas when $g_{ii}$ is increased, $x_i(t)$ decays sooner, leading to a reduction in $x_e(t)$ inhibition and subsequently an increase in $X_e(\omega)/X_i(\omega)$.

## PET data acquisition and image processing

Detailed descriptions of flortaucipir and PiB PET acquisition are available in previous publications (*Ossenkoppele et al., 2016*; *Schöll et al., 2016*). All PET scans were acquired at Lawrence Berkeley National Laboratory (LBNL) on Siemens Biograph 6 Truepoint PET/CT scanner (Siemens Medical Systems) in 3D acquisition mode. Flortaucipir was synthesized at the LBNL Biomedical Isotope Facility (BIF) using a GE TracerLab FXN-Pro synthesis module with a modified protocol based on an Avid Radiopharmaceuticals protocol supplied to the facility. Participants were injected with 10 mCi of tracer and scanned in listmode 80- to 100-min postinjection (4 × 5 min frames). [11]C-PIB was also synthesized at the LBNL BIF according to a previously published protocol (*Mathis et al., 2003*). Beginning at the start of an injection of 15 mCi of PIB into an antecubital vein, 90 min of dynamic emission data were acquired and subsequently binned into 35 frames (4 × 15, 8 × 30, 9 × 60, 2 × 180, 10 × 300, and 2 × 600 s). Flortaucipir and [11]C-PIB PET images were reconstructed using an ordered subset expectation maximization algorithm with weighted attenuation and smoothed with a 4-mm Gaussian kernel with scatter correction. Image resolution, calculated using a Hoffman brain phantom, was $6.5 \times 6.5 \times 7.25$ mm$^3$. Ninety minutes of dynamic postinjection data for PIB and 80- to 100-min postinjection data for flortaucipir were used for the following PET processing.

Each patient's MRI was segmented using Freesurfer 5.3 (http://surfer.nmr.mgh.harvard.edu; *Fischl et al., 2002*). PET data were realigned and coregistered onto their corresponding T1 image using the Statistical Parametric Mapping 12 (SPM12, http://www.fil.ion.ucl.ac.uk/spm/). SUVR images were created using Freesurfer-defined cerebellar gray matter for PIB-PET. For FTP, Freesurfer segmentation was combined with the SUIT template (*Diedrichsen, 2006*) to only include inferior cerebellum voxels therefore avoiding contamination from off-target binding in the dorsal cerebellum (*Baker et al., 2017*).

## Magnetic resonance image acquisition and analysis

Structural brain images were acquired from all participants using a unified MRI protocol on a 3 Tesla Siemens MRI scanner at the Neuroscience Imaging Center (NIC) at UCSF. Structural MRIs were used to generate individualized head models for source space reconstruction of MEG sensor data. The structural MRI scans were also used in the clinical evaluations of patients with AD to identify the pattern of gray matter volume loss to support the diagnosis of AD.

