## [Editor Report]

The authors explored the relationship between amyloid-β and tau deposition and neural oscillations in Alzheimer's disease (AD) by using a computational neural mass model that can generate neurophysiological power spectra comparable to EEG- or MEG-like, macroscopic brain activity assessments. This analysis demonstrates the different, frequency-specific effects of amyloid-β and tau proteins on excitation and inhibition, providing an integrated, multimodal explanation of AD pathogenesis.

---

## [Decision Letter]

**Decision letter after peer review:**

Thank you for submitting your article "Altered excitatory and inhibitory neuronal subpopulation parameters are distinctly associated with tau and amyloid in Alzheimer's disease" for consideration by *eLife*. Your article has been reviewed by 2 peer reviewers, and the evaluation has been overseen by a Reviewing Editor and Jeannie Chin as the Senior Editor. The following individual involved in the review of your submission has agreed to reveal their identity: Fernando Maestú (Reviewer #1).

Essential revisions:

1. In general, in the limitations section, one or more of the reported 'weaknesses' above could be discussed.

2. For the mediation analysis, the distinction between 'direct' effects and time-constant-mediated effects could be explained a bit more.

3. Was there any specific argument to concatenate the δ and theta bands, but not for example the α/β? Is this because of the expected spectral shape? This choice could perhaps be motivated a bit more elaborately.

4. If possible, a more in-depth explanation of the model parameters (e.g. what is neural gain?) could help the reader to better grasp the meaning of the observed changes.

5. While 12 subjects from a previous study (Ranasinghe et al., 2022 Brain) have been included, in which the presence/absence of subclinical epileptiform activity was identified, this has not been considered as a potential confounder. Also, in that study, functional connectivity appeared to be a better surrogate for excitability levels than spectral slowing. Would it be possible to investigate this with the present model? A reflection on these issues in the discussion would give more context.

6. The figures are very clear and easy to read. I would prefer to have the figure with the regional parameter changes (appendix figure 2) in the main text, as it enables a nice direct visual comparison between pathology and activity.

7. It would be fair to discuss the author's work in comparison to recent modelling work by Stefanovski et al., (https://doi.org/10.3389/fncom.2019.00054), which had a similar aim (although tau was not taken into consideration there).

8. Please include in the Discussion section some alternative explanations regarding the role of slow waves at this stage of the disease.

---

## [Author Response]

Essential revisions:1. In general, in the limitations section, one or more of the reported ‘weaknesses’ above could be discussed.

We have addressed all the concerns raised by reviwer-1 and the ‘weaknesses’ raised by reviwer-2. Please refer to the specific answers under each item from Reviewer #1 and #2.

2. For the mediation analysis, the distinction between ‘direct’ effects and time-constant-mediated effects could be explained a bit more.

Thank you for letting us explain this in depth. The associations between frequency-specific spectral changes and AD proteinopathy have been previously described by our group and others^1-5^. Specifically, increased low frequency δ-theta range oscillatory activity are strongly associated with increased amyloid depositions while reduced α oscillatory activity is strongly associated with increased tau depositions. In the current investigation, we demonstrated that higher amyloid is correlated with increased inhibitory time constants, whereas higher tau is correlated with increased excitatory time constants.

Mediation analyses described in this manuscript examined the contribution of time-constants to the relationship between oscillatory power changes and AD proteinopathy. In general, mediation analysis of a relationship between an independent and a depended variable quantify the mediated effects, the direct (un-mediated) effects and the overall effects which reflects the combination of the mediated and direct effects. In our case for example, we quantify the direct and overall effects between amyloid and δ-theta and those mediated by the inhibitory-time constant. We found that increased inhibitory time constant partially mediated the association between higher amyloid and increased δ-theta power, whereas increased excitatory time constant partially mediated the association between higher tau and reduced α power. Collectively, these findings suggest that altered time constants contribute to the signature changes in oscillations while AD proteinopathy also influence the spectral changes independent of neuronal parameter changes. We have now included these additional details into our methods and Results sections of the revised manuscript. (Page 9, lines 8-23)

3. Was there any specific argument to concatenate the δ and theta bands, but not for example the α/β? Is this because of the expected spectral shape? This choice could perhaps be motivated a bit more elaborately.

One of the signature spectral changes in patients with AD is an increased oscillatory power and synchrony in the low frequency spectrum. The peak of this low frequency spectrum occurs around 3-5 Hz which is bordering the conventional band cut-offs for δ (2-4Hz) and theta (4-8Hz) frequency oscillations. Therefore, to capture the low frequency spectral enhancement in its full strength and detail we combined the conventional δ and theta frequency bands to a single δ-theta ranging from 2-8 Hz. Furthermore, this approach help mitigate leakage effects of bandpass filtering in the narrower δ band. We have included the additional details of this rationale into the methods section of the revised paper. (Appendix 2: page 34, lines 15-20)

4. If possible, a more in-depth explanation of the model parameters (e.g. what is neural gain?) could help the reader to better grasp the meaning of the observed changes.

We use a linearized neural mass model for fitting observed regional spectra. This model includes gain and time-constant parameters for excitatory and inhibitory sub-populations that capture canonical circuit properties of regional neuronal activity. The excitatory and inhibitory time constant parameters characterize the duration of the neural responses (modelled by a Γ-shaped function) in each neuronal subpopulation respectively. It also characterizes the rate at which a local signal dissipates in absence of other inputs. Note that a lower time constant indicates a faster rate of change in signals while a higher time constant indicates a slower rate. The excitatory and inhibitory gain parameters correspond to the postsynaptic gain on the impulse response function of each neuronal subpopulation respectively. We have now included these additional details into the revised methods section of the manuscript. (Page 7, lines 6:11)

5. While 12 subjects from a previous study (Ranasinghe et al., 2022 Brain) have been included, in which the presence/absence of subclinical epileptiform activity was identified, this has not been considered as a potential confounder. Also, in that study, functional connectivity appeared to be a better surrogate for excitability levels than spectral slowing. Would it be possible to investigate this with the present model? A reflection on these issues in the discussion would give more context.

We would like to clarify that the subjects in the current study did not overlap with the study we published in 2022 in Brain,^6^ where we categorically identified the presence or absence of subclinical epileptiform activity in AD patients using a long-term (overnight) EEG protocol (LTM-EEG) and long-term simultaneous EEG and MEG sleep protocol. The study participants for the current investigation were not examined in LTM-EEG and long-term M/EEG sleep protocol and were not categorized for the presence of epileptiform activity. Although twelve subjects in the current study overlapped with our previous investigation published in early 2020 in the journal Science Translational Medicine^1^, this study also did not include LTM-EEG or MEG sleep protocols. Therefore, we are not in a position to correlate the current findings from the multimodal imaging and neural mass model parameter estimation to epileptiform activity. Nevertheless, the reviewer raises some interesting questions, and we currently have an ongoing investigation designed to explore these very associations that we hope to publish in the future.

6. The figures are very clear and easy to read. I would prefer to have the figure with the regional parameter changes (appendix figure 2) in the main text, as it enables a nice direct visual comparison between pathology and activity.

Thank you for this comment. We have included the original appendix Figure 2 into the main manuscript. (page 13, Figure.2)

7. It would be fair to discuss the author's work in comparison to recent modelling work by Stefanovski et al., (https://doi.org/10.3389/fncom.2019.00054), which had a similar aim (although tau was not taken into consideration there).

Thank you for this suggestion. We discuss the investigation by Stefanovski et. Al in our revised discussion of the manuscript. In brief Stefanovski et al., used a Virtual Brain Modeling platform and a Jansen-Rit model to simulate the slowing of EEG/LFP signals in AD. A key difference between our study and Stefavovski et. al study is that while ours is a linearized neural mass model (NMM), Stefavovski et al., used a non-linear form of NMM model. While linearizing is a simplification of the detailed underlying biophysics, recent comparisons among different models demonstrate that linear models sufficiently capture neuroimaging data with higher accuracy compared to non-linear models.^7^ In addition, the small set of model parameters and the closed-form solution in the frequency domain in our model makes the parameter inference more tractable compared to non-linear NMM. Indeed, we were able to show accurate fits to empirical spectra, capturing the empirical peak frequency as well as the frequency fall-off. However, with linearized models, we do not observe bifurcation points and other bi-stable behaviors that can be observed in a non-linear NMM. Nevertheless, despite methodological differences Stefanovski et al., also found a positive relationship between the inhibitory time constant and higher amyloid suggesting a relationship between amyloid accumulation and spectral slowing. This is consistent with our results where we have shown the regional level details of positive association between higher amyloid and increased inhibitory time constant in patients with AD using our local NMM approach. A novel finding in our study, as correctly pointed out by the reviewers, is that we also report a positive relationship between excitatory time constant and tau accumulation. We discuss the study by Stefanovski et al., in our revised discussion of the manuscript. (Page 22 Lines 4-25, Page 23, line 1-2)

8. Please include in the Discussion section some alternative explanations regarding the role of slow waves at this stage of the disease.

Thank you. We have now included a new figure summarizing our findings and discuss the relationship between slow waves and hyperexcitability (see also response to item number 4 by Reviewer #1) (Page 20, lines, 9-22)